# Effect of Herpes Zoster Treatment and Sudden Sensorineural Hearing Loss Using National Health Insurance Claims Data of South Korea

**DOI:** 10.3390/medicina59040808

**Published:** 2023-04-20

**Authors:** Hyo Jung Son, Eun-Ji Choi, Ukjin Jeong, Yoon Ji Choi

**Affiliations:** 1Department of Anesthesiology and Pain Medicine, National Police Hospital, Seoul 05715, Republic of Korea; 2Department of Dental Anesthesia and Pain Medicine, Dental Research Institute, School of Dentistry, Pusan National University, Yangsan 50612, Republic of Korea; 3Department of Anesthesiology and Pain Medicine, Korea University Ansan Hospital, Korea University College of Medicine, Ansan 15355, Republic of Korea

**Keywords:** geriatric population, herpes zoster, sudden sensorineural hearing loss

## Abstract

*Background and objectives*: Herpes zoster (HZ) is caused by the reactivation of a pre-existing latent varicella zoster virus, which is one of the viruses that causes hearing loss, and hearing loss may occur due to a systemic immune response even if it does not invade the auditory nerve. This study aimed to determine the correlation between sudden sensorineural hearing loss (SSNHL) in older adult patients who received HZ treatment. *Materials and Methods*: We used the cohort data of patients aged 60 years and above (*n* = 624,646) between 2002 and 2015 provided by the National Health Insurance Service. The patients were divided into two groups: those who were diagnosed with HZ between 2003 and 2008 (group H, *n* = 36,121) and those who had not been diagnosed with HZ between 2002 and 2015 (group C, *n* = 584,329). *Results*: In the main model (adjusted HR = 0.890, 95% CI = 0.839–0.944, *p* < 0.001) adjusted for sex, age, and income, and the full model (adjusted HR = 0.894, 95% CI = 0.843–0.949, *p* < 0.001) adjusted for all comorbidities, group H had a lower risk of SSNHL than group C. *Conclusions*: This study showed that patients who received HZ treatment had a lower incidence of SSNHL within five years after diagnosis.

## 1. Introduction

Herpes zoster (HZ) is a disease in which the varicella-zoster virus (VZV) causes chickenpox in childhood, and the virus does not disappear, even after treatment is complete, and remains dormant in the ganglion of the spinal cord, enters the nerve, and resumes activity when immunity is low [1,2]. It is characterized by a maculopapular or vesicular rash along the dermatome and is accompanied by severe pain. The most common complication of HZ is post-herpetic neuralgia (PHN), which is characterized by persistent pain and a skin rash that can negatively affect the patient’s quality of life, including physical, emotional, and social functioning.

The number of patients with HZ is increasing due to the growth in the older adult population. Additionally, the numbers of HZ neuralgia and other complications are equally increasing [3,4]. One such complication is sudden sensorineural hearing loss (SSNHL). VZV is a representative virus that can lead to varying degrees of hearing loss, typically SSNHL, and has been reported to occur in 6.5–8.5% [5]. VZV causes hearing loss through two main mechanisms: direct damage to the auditory nerve and inner ear structures, inner ear hair cells, and organ of Corti, and the induction of host immune-mediated damage [6]. Hearing loss caused by VZV is mostly due to the onset of herpes zoster oticus (HZO) caused by an infection of the eighth cranial nerve [7]. However, some patients develop hearing loss with an increase in Ig M for VZV, without evidence of auditory nerve involvement [8]. 

The cause or mechanism of idiopathic SSNHL is unknown, but if HZ is diagnosed and treated, antiviral or steroid treatment may affect the incidence of viral or idiopathic SSNHL. Therefore, we hypothesized that the incidence of hearing loss would be lower in patients treated for HZ with antiviral and/or steroidal therapy. A study that followed up on the incidence of SSNHL for two months in patients who had recently suffered from HZ reported no correlation between the two diseases. However, the follow-up period was too short to confirm the effect [9]. This study aimed to compare the incidence of SSNHL in patients diagnosed with and treated for HZ and the control group by analyzing the data from the National Health Insurance Service over 13 years. 

## 2. Materials and Methods

### 2.1. Study Population and Data Collection

This study was conducted in accordance with the Declaration of Helsinki and approved by the Institutional Review Board (Ansan, Republic of Korea; approval number 2020AS0009) of the Korea University Ansan Hospital.

The Korean government has implemented a compulsory National Health Insurance Service (NHIS) since 1989, which is a nationwide health care system that provides medical services for almost all of the country’s residents. All demographic data, including age and sex, are collected under the NHIS according to the patient’s Korean ID number. As the Health Insurance Review and Assessment Service (HIRA) was developed for medical billing purposes at the NHIS, all medical acts including diagnosis, physical and laboratory examination, treatment, prescription, nursing acts, and hospitalization are recorded relatively precisely in a computerized database along with the individual’s Korean ID number. Data collection was performed using the elderly cohort data provided by the NHIS between 2002 and 2015. The qualified individuals, those who were over the age of 60 (*n* = 624,646) in 2002, were provided. The cohort database included basic demographic information, socioeconomic data, medical comorbidities, diagnostic codes using the International Classification of Disease-10 (ICD-10), status of medical resource utilization (Consult and medical check-up), status of clinic and elderly long-term nursing service, and death records. The data were supplied after patient de-identification by the NHIS.

We included 624,646 patients who were aged 60 years and above in 2002 in this study. The exclusion criteria include patients who had already been diagnosed with HZ before 2002 or those who had died during the follow-up period (*n* = 4196). The patients were divided into two groups: those who had received the first diagnosis of HZ (ICD-10: B02) and received any treatment between 2003 and 2008 (group H, *n* = 36,121), and those without a diagnosis of HZ (group C, *n* = 584,329). We followed the medical records of the study participants for a minimum of 7 years and a maximum of 13 years. The occurrence of SSNHL in the patient was confirmed by ICD-10: H912. It is defined as the sudden onset of hearing loss of at least 30 dB in three consecutive frequencies in an audiometry test within 72 h. Among them, only patients who underwent an audiometry test were included. In group H, patients who had been diagnosed with SSNHL prior to a diagnosis of HZ or who had died between 2003 and 2008 were excluded (*n* = 669). In group C, those newly diagnosed with HZ between 2009 and 2015 were excluded (*n* = 97,751). Finally, 1:1 matching between the two groups was achieved according to age, sex, and income (*n* = 35,452), as shown in Figure 1. To avoid selection bias when matching the participants, the control group participants were sorted using a random number order and were randomly selected.

### 2.2. Statistical Analysis

The Pearson chi-square test and independent Student’s *t*-test were used to analyze the descriptive statistics and continuous variables to compare the sociodemographic characteristics and comorbidities of the two groups. The incidence rate was calculated as the number of SSNHL cases diagnosed during the follow-up period divided by the total number of life-years in each cohort for each follow-up period.

The Kaplan-Meier method was used to calculate the cumulative incidence of SSNHL between the two groups. The Log-Rank test was used to analyze the differences in the occurrence curves between the two groups. We used Poisson regression to estimate the incidence rate, which was used to compare the risk of developing SSNHL between the HZ and control group. The incidence of SSNHL according to the follow-up period was analyzed by dividing it into less than 1 year, between 1 and 5 years, and more than 5 years, according to the follow-up period.

The Cox-proportional hazards model was used to estimate the adjusted hazard ratio (HR) for SSNHL incidence. In this analysis, the main model (adjusted by age, sex, and income), full model (adjusted by age, sex, income, and comorbidities), and main with each additional listed comorbidity model were used. 

SAS^®^ ver. 9.4 (Statistical Analysis Software 9.4, SAS Institute Inc., Cary, NC, USA) and R software version 3.3.2 (R Project for Statistical Computing, Vienna, Austria) were used for all the statistical analysis. *p* values less than 0.05 were statistically significant. 

## 3. Results

Table 1 shows the characteristics of the study participants and the incidence of comorbidities in the elderly database. Among the 35,452 patients matched, group H showed a higher rate of SSNHL diagnosis. The incidence of SSNHL was 6.6% in group H and 6.0% in group C (*p* = 0.005). The rate of a diagnosis of comorbidities, such as myocardial infarction, cerebrovascular disease, rheumatic or connective disease (autoimmune disease), diabetes mellitus, moderate or severe renal failure, otitis media, and hypertension, was higher in group H than group C (*p* < 0.05).

Figure 2 shows the Kaplan-Meier survival curves and Log-Rank tests for the development of SSNHL in the elderly database. The cumulative incidence of SSNHL was significantly different between group C and group H (chi-square 23.356, DF 1, *p* < 0.001). The cumulative incidence of SSNHL was found to be significantly higher in patients without HZ (group C) than in patients with HZ (group H).

Table 2 shows the comparison of the mean SSNHL-free periods between group H and group C. The SSNHL free period was 8.703 ± 0.059 years in group H, which was longer than that in group C (7.773 ± 0.078 years).

Table 3 shows the incidence of SSNHL according to the follow-up period in the elderly database. A stratified analysis of the number of SSNHL cases by follow-up period showed that the cumulative incidence of SSNHL was significantly lower in group H than in group C within five years after the diagnosis of HZ (*p* < 0.005). However, Table 3 shows that the incidence of SSNHL was significantly higher in group H five years after the diagnosis of HZ (*p* < 0.001).

Table 4 shows the results of the multivariate Cox regression model analysis comparing the risk of SSNHL between the two groups. The incidence of SSNHL was significantly lower in group H than in group C. In the main model (adjusted HR = 0.890, 95% CI = 0.839–0.944, *p* < 0.001) adjusted for sex, age, and income, and the full model (adjusted HR = 0.894, 95% CI = 0.843–0.949, *p* < 0.001) adjusted for all comorbidities, group H had a lower risk of SSNHL than group C. Similar results were found in the sensitivity analysis of the potential confounding variables caused by each comorbidity (myocardial infarction, cerebrovascular disease, rheumatic or connective disease, diabetes mellitus, moderate or severe renal failure, otitis media, and hypertension). Gender stratification analysis showed that women had a significantly lower risk of developing SSNHL in group H than in group C (HR = 0.880, 95% CI = 0.820–0.944, *p* < 0.001). Age stratification analysis showed a significant association between the incidence of HZ and SSNHL in patients aged 70–80 years and those above 80 years (HR = 0.919, 95% CI = 0.850–0.994, *p* = 0.034 and HR = 0.862, 95% CI = 0.788–0.944, *p* = 0.001).

## 4. Discussion

In the elderly database, the incidence of SSNHL in HZ patients was higher than in group C and the cumulative incidence of SSNHL was found to be significantly higher in patients without HZ (group C) than in patients with HZ (group H). However, in the multivariable Cox regression model adjusted for comorbidities, there was a significantly lower incidence of SSNHL in group H than in group C, and the risk of SSNHL was significantly lower in females and patients above 70 years.

HZ, also known as shingles, is characterized by a blistering rash that occurs in bands along the dermatome and causes severe pain [10,11]. Depending on the nerve segment involved in the HZ, various complications are also induced. PHN, which is the most common complication, causes excruciating pain that significantly reduces patients’ quality of life, causing functional physical impairment, emotional distress, and interfering with daily activities and sleep [12]. In addition, there are complications, such as secondary bacterial infection in the healing process of the blistering rash, HZ ophthalmicus with eye involvement, neurological sequelae, and segmental paresis affecting muscles. 

The lifetime expected risk of being diagnosed with HZ in the general population is approximately 30%, and the risk increases rapidly after age 50, up to 50% for those over 80 years [13]. There is also an increased risk of the complications associated with HZ, particularly long-lasting PHN and hospitalization in the elderly population [14]. However, it is not a disease that should be noted only in the elderly population as HZ patients may spread the virus to seronegative contacts for VZV and they may develop varicella, but not HZ [15,16]. Even in the case of immunocompetent individuals, more than 95% of those over 50 years of age are seropositive, and the rate of household transmission of HZ is 15%, so contact transmission is sufficiently dangerous [17].

SSNHL is a sudden onset of sensorineural hearing loss ≥ 30 dB or more, at least three contiguous audiometric frequencies, within 72 h [18]. The prevalence of SSNHL is 1.5–1.7% of the total population, and among these, only 30% or less of patients know the exact cause and receive appropriate treatment [19,20,21]. More than half of the patients are classified as idiopathic SSNHL [18,22,23]. 

When summarized into broad categories, the known causes of SSNHL include infectious, autoimmune, metabolic, neoplastic, neurologic, otologic, toxic, and traumatic [24]. Among these, infection (13%) was the most frequently identified cause. The viral infections known to cause SSNHL include cytomegalovirus, rubella, measles, mumps, human immunodeficiency virus, herpes simplex virus, VZV, and West Nile virus [5]. There are three major mechanisms by which viral infections cause SSNHL [25,26]. First, through the direct invasion of the virus into the fluid space and/or soft tissue of the cochlear(cochleitis) or cochlear nerve (neuritis). The virus infects the inner ear mainly through hematogenous transmission, but infection through the cerebrospinal fluid space or the middle ear is possible. Second, the virus can infect the inner ear through the reactivation of the dormant virus under conditions of significant stress [27]. Third, it is explained by an indirect mechanism of immune-system-mediated action against the virus. According to this hypothesis, systemic or distant viral infections trigger an antibody response that cross-reacts with an inner ear antigen [6,28,29,30,31]. Antibody titers against several viruses are seroconverted in patients with idiopathic SSNHL [32,33]. When the histopathological findings of the temporal bone were investigated in patients who had experienced SSNHL at least once in their lifetime, many patients had no trace of a direct viral infection in the inner ear. This supports an indirect mechanism of the virus [26]. As SSNHL is more likely to recover the earlier it is treated, early detection and treatment are very important in emerging diseases. Treatment is applied in various ways depending on the cause, but the required treatment can be difficult to determine because there are many cases of idiopathic SSNHL. 

In general, the treatment includes the use of corticosteroids, which are administered through oral medication or injection behind the eardrum [10,14]. SSNHL may recover spontaneously, but with the appropriate treatment within one-to-two weeks of the onset of symptoms, more than 50% of patients achieve complete or partial hearing recovery [11]. A poor prognosis is defined as when the hearing loss is severe, with high-pitched tone impairment and full frequency range loss, speech intelligibility is reduced, vertigo is present, in children and over 40 years of age, and with the late initiation of treatment [13].

VZV is a double-stranded enveloped DNA virus that belongs to the Herpesviridae family. VZV is a highly contagious virus transmitted by air droplets when an infected person coughs or sneezes, or through direct contact with herpetic vesicle exudates [5]. Some patients are asymptomatic at the initial infection with VZV, and when symptomatic, it causes fever, erythematous macular rash, and pustules (chickenpox). Even after treatment, the virus remains in the neurons of various body parts and reactivates several years later. The symptoms of VZV reactivation include localized pain and paresthesia, red spots, blisters along the area dominated by the sensory nerves of the neuromuscular root, and systemic symptoms, such as fever and fatigue [1,34]. 

Patients with T-lymphocyte-macrophage-mediated immune system deficiency have a more severe viral spread, which can cause complications in the visceral or nervous system beyond the ganglion, nerve, and dermatome units [3,11]. The goals of treatment are to reduce the spread, duration, and severity of the initial infection, prevent the spread to other parts of the body, control pain in the acute phase, and prevent later progression to chronic PHN. In particular, early treatment is very important in patients with a high frequency of complications of HZ, such as elderly patients or immunosuppressed patients. The appropriate use of antiviral drugs and active pain management at an early stage are helpful in treating acute HZ and preventing complications [1,28]. The administration of antiviral drugs within 72 h of the diagnosis of HZ is the standard treatment for HZ in the acute phase. This agent has the effect of inhibiting viral replication and shortening the spread period, promoting the healing of rashes and reducing the duration and severity of acute pain. However, in many cases, acute pain control and the prevention of complications cannot be solved with antiviral drugs alone; therefore, many patients’ treatment is accompanied by additional drugs or procedures. Additional prescribed oral medications include non-opioid analgesics such as NSAIDs, acetaminophen, or tramadol, and corticosteroids, anticonvulsants, and antidepressants may also be used. When oral drug treatment does not achieve a sufficient effect on pain control, a nerve block is performed, in which case corticosteroids are often used along with a local anesthetic. 

Ramsay Hunt syndrome, or HZO, is caused by the reactivation of latent VZV in the geniculate ganglion [7,35,36]. The virus enters the 8th cranial nerve through the internal auditory canal after being transmitted from the nearby geniculate ganglion. SSNHL is uncommon without evidence of nerve involvement in the ear [36]. There is also a report that people who repeatedly suffer from Ramsay Hunt syndrome have an autoimmune system problem [31]. Unfortunately, no reliable serological tests are available to diagnose reactivation. Once a patient is seropositive for a latent herpes virus, an increase in antibody titer does not diagnose reactivation. 

Several previous studies have reported no association between idiopathic SSNHL and HZV infection [9,37]. However, there were several limitations in the previously published studies. After being diagnosed with HZ, the follow-up period was only three months [9], or patients who received steroid treatment with HZ oticus were matched [29]. When matching the two groups, it is also important to manage underlying diseases that may affect the patient’s immunity. Patients with HZ are often in a reduced state of immunity due to the general deterioration of their physical condition before the onset of symptoms, similar to those with SSNHL. Although an individual’s acute stressful situation can rapidly reduce their immunity, the underlying disease or drugs used to treat HZ can have a long-term effect on the patient’s immunity. 

In this study, we analyzed the long-term follow-up data from a minimum of seven to a maximum of 13 years. Considering that the HZ treatment effect may be short-term, the follow-up period was analyzed by dividing it into less than one year, between one and five years, and more than five years. In addition, multivariate analysis was also performed, including various underlying diseases that could affect the results. The therapeutic role of corticosteroids and/or corticosteroids with antiviral agents for SSNHL has not been fully proven yet, although corticosteroids are known to alleviate hearing loss and reduce cochlear damage in patients receiving this treatment by strongly suppressing immune-mediated inflammatory reactions [32,38,39]. Acyclovir and valacyclovir, two anti-herpes virus medications commonly used to treat HZ, are efficacious only against HSV and VZV, but have not been proven to be efficacious against SSNHL and may cause ototoxicity [32,38,40,41,42]. Although the first two drugs did not show a clear effect on short-term hearing restoration after a diagnosis of SSNHL, they may have reduced the long-term course or deterioration. 

The live attenuated vaccines against HZ are used worldwide and are recommended for people over the age of 50 with an increasing incidence of HZ. It has been reported that the HZ prevention effect is about 53% when targeting the population aged 60 years or older, and the morbidity of chronic PHN is reduced in 63% of cases [43].

This study analyzed a large national population using reliable data provided by the National Health Insurance Service of Korea. As the entire population is obligated to subscribe to the national health insurance and it is managed in Korea, individual medical data are not omitted or duplicated, and the records are thoroughly coded. Although the data are reliable, there are some limitations to this study. First, as this study is a retrospective cohort study, it is difficult to explain the exact mechanism by which the factors related to HZ treatment reduce the incidence of SSNHL. This is because there is a commonly used treatment protocol, but not all patients receive the same treatment. Second, many underlying diseases were considered and applied when matching the control group. However, some diseases that are not common were not considered. Third, cases of spontaneously resolved SSNHL may have been missed in this study population. In our data, the incidence of SSNHL was approximately 6% in patients over 60 years old, which was approximately 0.5% when converted to the total population. In previous studies, the incidence of SSNHL was 1.5–1.7%. However, our study showed a low incidence. This could be due to differences in the prevalence rates by country and race. Fourth, our populations have many comorbidities; therefore, many of these patients received a lot of medication for various comorbidities during this five-year follow-up period, which may influence the incidence of SSNHL. Therefore, we included comorbidities (diseases described in Table 1) that had a high frequency during the entire study period in the multivariate analysis, but these may affect the results.

## 5. Conclusions

In conclusion, the incidence of SSNHL in patients with HZ is reduced for up to five years after diagnosis. According to our findings, a series of HZ treatments or an acquired immunity to VZV influenced the reduction in the incidence of SSNHL within five years of HZ diagnosis. As our study was a retrospective cohort study, prospective randomized control studies to confirm our findings on the effect of HZ on SSNHL are recommended in the future.

## Figures and Tables

**Figure 1 medicina-59-00808-f001:**
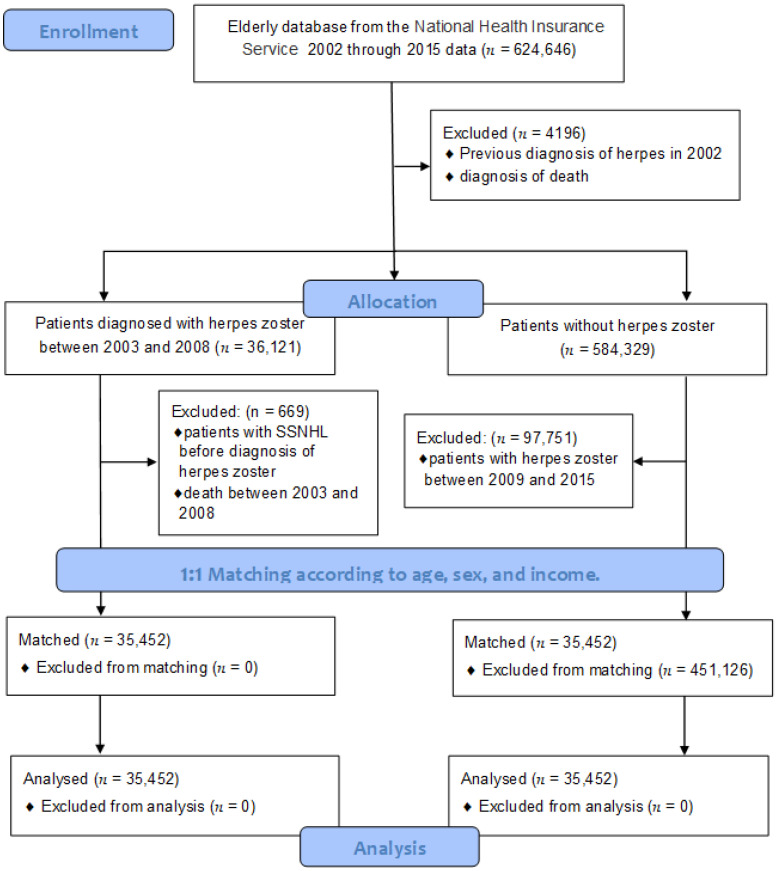
Flow diagram of the patient selection process in this study.

**Figure 2 medicina-59-00808-f002:**
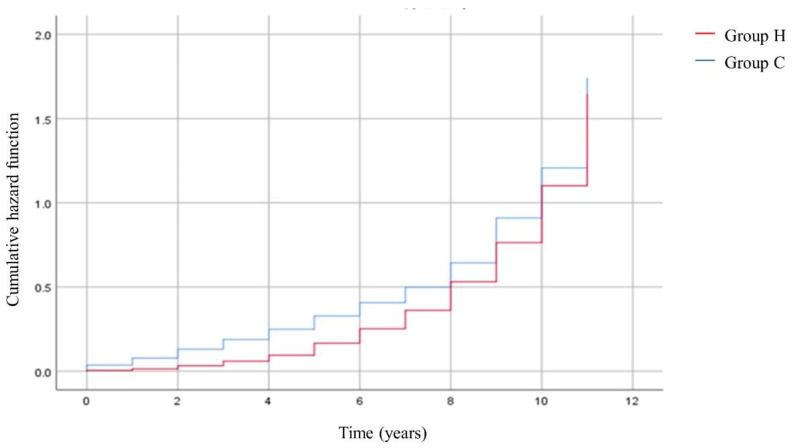
Kaplan-Meier survival curves and Log-Rank tests for development of SSNHL.

**Table 1 medicina-59-00808-t001:** Characteristics of the study participants and incidence of comorbidities.

	Group H (*n* = 35,452)	Group C (n = 35,452)	*p* Value
Age (years)	79.50 ± 5.90	79.50 ± 5.90	0.999
Sex			
M	13,500 (38.1)	13,500 (38.1)	1.000
F	21,952 (61.9)	21,952 (61.9)	
Income			
Low	4285 (12.1)	4285 (12.1)	1.000
Middle	16,316 (46.0)	16,316 (46.0)	
High	14,851 (41.9)	14,851 (41.9)	
Comorbidities			
Myocardial infarct	2470 (7.0)	2224 (6.3)	
Cerebrovascular disease	16,545 (46.7)	15,005 (42.3)	<0.001
Rheumatic or connective disease (autoimmune disease)	8163 (23.0)	6432 (18.1)	<0.001
Diabetes mellitus	16,693 (47.1)	14,200 (40.1)	<0.001
Moderate or severe renal failure (chronic renal disease)	594 (1.7)	509 (1.4)	0.010
Otitis media	8887 (25.1)	6965 (19.6)	<0.001
Hypertension	28,688 (80.9)	26,509 (74.8)	<0.001
SSNHL	2325 (6.6)	2143 (6.0)	0.005

Values are mean ± SD or number (percent). SSNHL: sudden sensorineural hearing loss. *p* < 0.05 compared between groups.

**Table 2 medicina-59-00808-t002:** Comparison of mean SSNHL free period.

Group	Average (Year)	SE	95% CI	
			Lower	Upper
Group H	8.703	0.059	8.588	8.818
Group C	7.773	0.078	7.620	7.927
Overall	8.257	0.049	8.161	8.353

**Table 3 medicina-59-00808-t003:** Incidence of SSNHL according to follow-up period.

	Group H				Group C				IRR	*p* Value
	*n*	SSNHL	Person-Years	Rate *	*n*	SSHL	Person-Years	Rate *		
overall	35,452	2325	2,715,540	8.6	35,452	2143	2,415,324	8.9	1.085 (1.023–1.151)	0.006
Year of follow-up										
<1	35,452	11	35,452	3.1	35,452	75	35,452	21.2	0.147 (0.078–0.276)	0.002
1–5	35,390	345	689,031	5.0	34,326	523	630,980	8.3	0.660 (0.576–0.756)	0.000
5<	32,582	1969	1,991,057	9.8	28,875	1545	1,748,892	8.8	1.274 (1.192–1.362)	0.000

Values are number. Rate *: per 10,000 person-years; IRR: Incidence rate ratio. The incidence rate ratio was compared using Poisson regression.

**Table 4 medicina-59-00808-t004:** Risk of SSNHL compared by multivariable Cox regression model.

Variable	Adjusted HR	95% CI		*p* Value
		Low	high	
Main model	0.890	0.839	0.944	<0.001
Full model	0.894	0.843	0.949	<0.001
Additional covariates				
Main model + Myocardial infarct	0.890	0.839	0.944	<0.001
Main model + Cerebrovascular disease	0.889	0.838	0.943	<0.001
Main model + Rheumatic or connective disease (autoimmune disease)	0.891	0.840	0.945	<0.001
Main model + diabetes mellitus	0.891	0.841	0.945	<0.001
Main model + Moderate or severe renal failure (chronic renal disease	0.889	0.839	0.943	<0.001
Main model + Otitis media	0.894	0.843	0.948	<0.001
Main model + hypertension	0.890	0.840	0.944	<0.001
Subgroup effects *				
Sex				
M	0.933	0.837	1.040	0.210
F	0.880	0.820	0.944	<.001
Age				
60–70	0.289	0.360	2.307	0.241
70–80	0.919	0.850	0.994	0.034
80-	0.862	0.788	0.944	0.001

Values are number. HR, hazard ratio; CI, confidence interval. The main model was adjusted for sex and age. The full model was adjusted for sex, age, and comorbidities. Subgroup effects *: adjusted for sex, age, and comorbidities.

## Data Availability

The data are unavailable due to the policy of the National Health Insurance Service.

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
