# Peer review of "Effect of Herpes Zoster Treatment and Sudden Sensorineural Hearing Loss Using National Health Insurance Claims Data of South Korea"

_medicina, 2023, doi:10.3390/medicina59040808_

Round 1

Reviewer 1 Report

In this study is suggested that patients who received HZ treatment had a lower incidence of SSNHL within 5 years after diagnosis. It is thought that appropriate treatment of HZ can potentially prevent SSNHL.

Medication for HZ treatment has no long-term effect to protect a patient from HZ recurrence or SSNHL due to the onset of herpes zoster oticus.

The authors should explain the mechanism by which these drugs act on long-term protection from SSNHL.

The authors should mention the treatment protocol for HZ, and if it was the same for all patients.

Many of these patients received a lot of medication for various comorbidities during this five years follow-up period which may influence the incidence of SSNHL.

Author Response

In this study is suggested that patients who received HZ treatment had a lower incidence of SSNHL within 5 years after diagnosis. It is thought that appropriate treatment of HZ can potentially prevent SSNHL.

Medication for HZ treatment has no long-term effect to protect a patient from HZ recurrence or SSNHL due to the onset of herpes zoster oticus.

The authors should explain the mechanism by which these drugs act on long-term protection from SSNHL.

Response: Thanks for your kind reminders. Since our study is a retrospective cohort study, there is a disadvantage that we cannot assert the exact mechanism for the statistical results. Regarding the part you commented on, I have additionally described it in detail in the discussion part as a limitation of the study.

The authors should mention the treatment protocol for HZ, and if it was the same for all patients.

Response: Thanks for your nice reminder. The general treatment protocol received by patients diagnosed with HZ is further described in the discussion section. In general, antiviral treatment is given, and if the pain is severe, nerve block is performed along with oral pain relievers. However, the use of antiviral drugs or other additional treatments may not have been matched with the same treatment between the two groups, so this point was added as a limitation of the study.

Many of these patients received a lot of medication for various comorbidities during this five years follow-up period which may influence the incidence of SSNHL.

Response: Thank you very much for the reminder. We included comorbidities (diseases described in table 1) with high frequency during the entire study period in multivariate analysis, so we think the influence of each medication difference on the statistical results will be small. We also added as a limitation of the study.

Reviewer 2 Report

First, I want to thank the Editor for the chance to review this interesting work. The aim of this work was to analyze the potential correlation between 15 sudden sensorineural hearing loss in older adult patients who received HZ treatment

“Patients with HZ are often immunocompromised before symptoms onset, similar to those with SSNHL”. Not only immunocompromised subjects experience SSNHL. Explain this sentence.

The authors wrote that “Patients who received HZ treatment had a lower incidence of SSNHL within 5 years after diagnosis. It is thought that appropriate treatment of HZ can potentially prevent SSNHL.” The  main treatment for HZV infection is the use of antivirals which, according to the last American guidelines, are not recommended for the treatment of SSNHL. So, how could it be the therapeutic mechanism of antivirals for prevent SSNHL in HZV treated patients?

The retrospective nature of the study should be included in the limits.

The incidence of comorbidities between group H and group C is statistically different, so this could explain the different correlation with SSNHL in the two groups.

The English language needs to be reviewed, and there are many errors to correct throughout the text (for example, not “sensoryneural” but “sensorineural”)

Author Response

Response to Reviewer 2

First, I want to thank the Editor for the chance to review this interesting work. The aim of this work was to analyze the potential correlation between 15 sudden sensorineural hearing loss in older adult patients who received HZ treatment.

Response: Thank you very much for your previous comments that helped us improve this manuscript.

“Patients with HZ are often immunocompromised before symptoms onset, similar to those with SSNHL”. Not only immunocompromised subjects experience SSNHL. Explain this sentence.

Response: Thanks for your nice reminder. We tried to explain the reduced state of immunity due to general deterioration of physical condition, but the expression in the text was modified because it could be misunderstood as occurring in immunocompromised patients.

The authors wrote that “Patients who received HZ treatment had a lower incidence of SSNHL within 5 years after diagnosis. It is thought that appropriate treatment of HZ can potentially prevent SSNHL.” The main treatment for HZV infection is the use of antivirals which, according to the last American guidelines, are not recommended for the treatment of SSNHL. So, how could it be the therapeutic mechanism of antivirals for prevent SSNHL in HZV treated patients?

Response: Thanks for your thoughtful question. Since our study is a retrospective cohort study, there is a disadvantage that we cannot assert the exact mechanism for the statistical results. Therefore, we did not conclude that the result of the low incidence of SSNHL in patients who received HZ treatment during the 5-year follow-up period after diagnosis was due to the effect of any treatment used on the patient. This is an area where prospective research should be conducted in the future, and as a limitation of our study, it was additionally described in the discussion section.

The retrospective nature of the study should be included in the limits.

Response: Thanks for your kind reminder. Additional limitations have been described in the discussion section.

The incidence of comorbidities between group H and group C is statistically different, so this could explain the different correlation with SSNHL in the two groups.

Response: Thanks for your nice reminder. We included comorbidities (diseases described in table 1) with high frequency during the entire study period in multivariate analysis, so we think it's statistically well represented. We also added as a limitation of the study.

The English language needs to be reviewed, and there are many errors to correct throughout the text (for example, not “sensoryneural” but “sensorineural”)

Response: Thanks for your kind reminder. We checked the English expression errors as a whole, and in particular, we thoroughly corrected the parts you pointed out.

Reviewer 3 Report

Thank you for the opportunity to review the article.
Title: Effect of herpes zoster treatment and sudden sensorineural hearing loss using national health insurance claims data of South Korea

Reviewer’s comments below:
Abstract
Has captured the study well.
Edit. Line 19: replace herpes zoster with HZ
Introduction
The introduction is concise and clearly presented.
The hypothesis may need a bit of re-wording to improve clarity. Line 44 – 45: “… would be lower in patients treated with herpes zoster, including antiviral therapy.” Amended to: …patients treated for herpes zoster with antiviral therapy.
Methods
Systematically presented, however, to improve more clarity, for the authors to consider the following suggestions:
Figure 1 – please improve quality of the image. Needs to be properly edited. Patients without HZ, not so clearly how the authors calculated their final matched from 584 329 to 35 452.
Line 58: suggestion – to include “criteria for severity/degree of SSNHL”
Results
Line 94: “The average survival period”. What does this mean? Is it the average survival period after HZ diagnosis?
Line 100: “…that the incidence of…” Suggestion to include “the cumulative incidence…”
Table 4: “ariable” – variable
Line 125: “…and SSNHL in patients aged 70-80 years… P = 0.034…” Please reconsider significance here.
Discussion
Please consider the following suggestions:
In Table 3, <1year. Please include the implications of this finding in the discussion section.
Line 133 – 134: “SSNHL is a sudden onset of sensorineural hearing loss ≥ 30dB or more at least three 133 contiguous audiometric frequencies within 72 hours.” Please include discussions relevant for your data and findings, that is, around SSNHL onset and cumulative analysis in view of this definition.
Line 134: “The prevalence is 1.5-1.7%...” The is word missing “ The prevalence of… is 1.5…”
Line 173 – 175: “However, these studies were limited to comparing incidence rates 173 over a short follow-up period or matching with controls for an underlying disease that 174 could affect the patient's immunity
was not properly managed.” Please consider commenting on the study findings and relevant for this statement.
Other comments
For the authors to consider improving the discussion section to focus it more to the study findings.

Author Response

Thank you for the opportunity to review the article.

Title: Effect of herpes zoster treatment and sudden sensorineural hearing loss using national health insurance claims data of South Korea

Response: Thank you very much for your previous comments that helped us improve this manuscript.

Reviewer’s comments below:

Abstract
Has captured the study well.
Edit. Line 19: replace herpes zoster with HZ

Response: Thanks for your kind reminder. We have corrected the part you pointed out.

Introduction

The introduction is concise and clearly presented.

The hypothesis may need a bit of re-wording to improve clarity. Line 44 – 45: “… would be lower in patients treated with herpes zoster, including antiviral therapy.” Amended to: …patients treated for herpes zoster with antiviral therapy.

Response: Thanks for your kind reminder. However, since antiviral drugs are not the only drugs used for HZ treatment, we modified the expression to “treated for HZ with antiviral and/or steroidal therapy”.

Methods

Systematically presented, however, to improve more clarity, for the authors to consider the following suggestions:

Figure 1 – please improve quality of the image. Needs to be properly edited. Patients without HZ, not so clearly how the authors calculated their final matched from 584 329 to 35 452.

Response: Thanks for your precise advice. It was confirmed and corrected that quality of the image was lowered in the process of attaching the figure file. The figure was modified by adding that 1:1 matching between the two groups was performed, and as a result, it was confirmed and corrected that the number of excluded n was incorrectly entered as 0 (n = 451,126). In addition, the excluded data are described in the data collection section of the text.

Line 58: suggestion – to include “criteria for severity/degree of SSNHL”

Response: Thanks for your advice. Additional explanations about the diagnostic criteria for SSNHL (ICD-10) have been added. However, no classification was made according to severity in data collection.

Results
Line 94: “The average survival period”. What does this mean? Is it the average survival period after HZ diagnosis?

Response: Thanks for your kind reminder. I correct the words.

Line 100: “…that the incidence of…” Suggestion to include “the cumulative incidence…”

Table 4: “ariable” – variable

Response: Thanks for your kind reminder. We have corrected both as you pointed out.

Line 125: “…and SSNHL in patients aged 70-80 years… P = 0.034…” Please reconsider significance here.

Response: Thanks for your advice. Sorry, I didn't understand the question well. Could you please explain in more detail?

Discussion

Please consider the following suggestions:

In Table 3, <1year. Please include the implications of this finding in the discussion section.

Response: Thanks for your kind reminder. Statistical analysis was conducted by dividing the follow-up period in consideration of the sustained effect of HZ treatment or the antibody generated in the patient. If treatment for HZ affects the results of the study, we expected that the period within 1 year of treatment would be most affected, and divided the period that way, resulting in similar results as within 5 years. In the discussion section, we have additionally described the reasons for dividing the data into less than 1 year, 1 to 5 years, and more than 5 years after HZ in the statistical analysis.

Line 133 – 134: “SSNHL is a sudden onset of sensorineural hearing loss ≥ 30dB or more at least three 133 contiguous audiometric frequencies within 72 hours.” Please include discussions relevant for your data and findings, that is, around SSNHL onset and cumulative analysis in view of this definition.

Response: Thanks for your precise advice. Since our study is a retrospective cohort study, it was not possible to perform detailed progression classification other than the classification of patients according to the ICD-10: B912 code in collecting data. Data were collected by checking the patient's diagnosis code and checking whether the audiometry test was performed.

Line 134: “The prevalence is 1.5-1.7%...” The is word missing “The prevalence of… is 1.5…”

Response: Thanks for your kind reminder. We have corrected it as you pointed out.

Line 173 – 175: “However, these studies were limited to comparing incidence rates 173 over a short follow-up period or matching with controls for an underlying disease that 174 could affect the patient's immunity

was not properly managed.” Please consider commenting on the study findings and relevant for this statement.

Response: Thanks for your careful comment. We have tried to explain the limitations of the cited paper more neatly.

Other comments

For the authors to consider improving the discussion section to focus it more to the study findings.

Response: Thank you very much for the reminder. Overall, the discussion section was revised to reorganize the meaning and limitations of the research results.